# A longitudinal analysis of selective motor control during gait in individuals with cerebral palsy and the relation to gait deviations

**Gilad Sorek**[1], **Marije Goudriaan**[2,3], **Itai Schurr**[4], **Simon-Henri Schless**[1,4]*

**1** Laboratory for Paediatric Motion Analysis and Biofeedback Rehabilitation, ALYN Helmsley Paediatric and Adolescent Rehabilitation Research Centre, Jerusalem, Israel, **2** Department of Human Movement Sciences, Vrije Universiteit Amsterdam, Amsterdam, The Netherlands, **3** Department of Rehabilitation Medicine, Amsterdam UMC, Amsterdam, Netherlands, **4** Clinical Motion Analysis Laboratory, ALYN Paediatric and Adolescent Rehabilitation Centre, Jerusalem, Israel

* shschless@alyn.org, simon.h.schless@gmail.com

**Data Availability Statement:** All relevant data are available on Figshare: 10.6084/m9.figshare.23691660.

## Abstract

### Objective

To investigate longitudinal changes in selective motor control during gait (SMCg) in individuals with cerebral palsy (CP), and to assess if they are related to changes in gait deviations.

### Method

Twenty-three children/adolescents with spastic CP (mean ± SD age = 9.0±2.5 years) and two 3D gait assessments (separated by 590±202 days) with no interim surgical intervention, were included. SMCg was assessed using muscle synergy analysis to determine the dynamic motor control index (walk-DMC). Gait deviation was assessed using the Gait profile score (GPS) and Gait variable scores (GVS).

### Results

There were no mean changes in walk-DMC score, GPS or GVS between assessments. However, changes in walk-DMC scores in the more involved leg related to changes in hip flexion-extension and hip internal-external GVS ($r_p$ = -0.56; p = 0.017 and $r_p$ = 0.65; p = 0.004, respectively).

### Conclusions

On average, there were no significant longitudinal changes in SMCg. However, there was considerable variability between individuals, which may relate to changes in hip joint kinematics. This suggests that a combination of neural capacity and biomechanical factors influence lower limb muscle co-activation in individuals with CP, with a potential important role for the hip muscles. These findings highlight the importance of taking an individualized approach when evaluating SMCg in individuals with CP.

**Funding:** SHS is a principal investigator and GS is a research fellow, both funded by The Leona M. & Henry B. Helmsley Charitable Trust (no. 2207-05386). SHS also received internal funding from ALYN hospital supported by the Centre for Integration in Science at the Ministry of Absorption and integration. MG was funded by the Dutch Organization for Scientific Research (NWO) VIDI grant (no. 016.156.346 FirSTeps), the European Research Council (ERC) under the European Union's Horizon 2020 research and innovation program (no. 715945 Learn2Walk), the Johanna Kinderfonds and Kinderrevalidatie Fonds Adriaanstichting (no.20200028), and internal funding of the Vrije Universiteit Amsterdam, the Netherlands. The funders had no role in study design, data collection and analysis, decision to publish, or preparation of the manuscript.

**Competing interests:** The authors have declared that no competing interests exist.

# Introduction

Cerebral palsy (CP) is the most common cause of childhood physical disability [1]. The clinical manifestations of CP depend on the level of motor development, type of movement disorder and the affected parts of the body [1]. Individuals with CP often present with changes such as bone deformity [2], abnormal muscle tone [3], decreased joint range of motion (ROM) [4, 5], muscle weakness [1] and atypical selective motor control [6, 7]. Previous studies have highlighted that selective motor control may be one of the more meaningful clinical symptoms with respect to the development of pathological gait patterns [6].

During the last decade, advanced techniques, such as muscle synergy analysis, have been increasingly used to quantify selective motor control during gait (SMCg) [6]. The term muscle synergies refers to groups of muscles that are activated simultaneously during rhythmic tasks, such as walking. They consist of synergistic activations (e.g., timing of muscle activity during a gait cycle) and the weighted contribution of each muscle to the synergy activations (synergy weights). Each synergy usually contains multiple muscles and each muscle can contribute to multiple synergies [6, 8, 9].

In typically developing (TD) individuals, two lower extremity muscle synergy patterns (locomotor primitives) are present in neonates during stepping movements [10]. With more walking experience, the number of synergies increases to four, with each synergy corresponding to a specific event in the gait cycle [10]. Previous research in TD children revealed that SMCg matures during the early developmental period when learning to walk [10, 11]. Once independent walking is achieved, the complexity of SMCg appears to remain stable [12, 13].

Due to delayed and altered development of the corticospinal tracts, individuals with CP present with atypical SMCg, leading to an impaired ability to coordinate complex lower limb muscle patterns. They often present with more muscle co-activation, which is reflected in fewer muscle synergies during gait (e.g., decreased SMCg) compared to TD children [6, 12, 13]. Decreased SMCg was also found to relate to reduced gait function and clinical outcome measures such as spasticity and muscle weakness [12]. Previous studies have shown that SMCg is an important factor in anticipating treatment outcomes in individuals with CP; individuals with more complex SMCg demonstrated greater improvements in walking speed and gait deviations after treatment, regardless of conservative or surgical intervention [14, 15]. However, on the group level, there were only minimal changes in SMCg after surgical interventions, despite changes in walking patterns [16].

Previous studies investigating the natural development of SMCg in individuals with CP used either a cross-sectional analysis research design [12, 13] or a few case studies of young individuals with CP around the time of achieving independent walking, with a follow-up period of one to two years [11]. How SMCg naturally develops in older children with CP over a longer period of time remains unknown. In addition, there is evidence that afferent input from the lower extremities (e.g. hip position and foot sole pressure) plays an important role in the development of motor control of walking [17, 18]. Hence, it may be that walking experience and longitudinal changes in gait deviations [19, 20], such as joints kinematics, may influence SMCg.

The aim of this study was to investigate longitudinal changes in SMCg in a group of individuals with CP, after achieving the milestone of independent ambulation, and to determine whether this relates to changes in gait deviations. The first hypothesis is that despite an expected high variability in SMCg [6, 16, 21], there will be no significant change in this measure between assessments. The second hypothesis states that there will be significant relationships between changes in measures of SMCg and gait deviations [11].

## Materials and methods

### Participants

This study was based on a retrospective convenience sample of 23 participants, evaluated as part of their clinical management in the ALYN motion analysis laboratory, Jerusalem, Israel, from January 1, 2019 to September 30, 2022. Individuals were referred for a clinical gait analysis to determine the benefit of monitoring gait function to improve: 1) orthopedic management, 2) spasticity management, 3) effect of ankle foot orthoses, 4) assistive devices and 5) physiotherapy treatment.

The inclusion criteria were 1) diagnosis of spastic CP with unilateral or bilateral lower limb involvement, 2) Gross Motor Function Classification System (GMFCS) level I-III with the ability to walk on a treadmill with or without handrails, 3) at least six months since previous botulinum injection, 12 months since previous orthopaedic intervention and two years since a selective dorsal rhizotomy, and 4) two gait assessments with an interim period of at least six months that did not include an invasive intervention (botulinum injection or surgery).

In addition, a convenience sample of 16 TD children were included to provide control reference values (Table 1). The main outcome parameters of this study have been shown to be stable between the age of 4–18 [12, 22]. As such, we do not consider the heterogeneous characteristics of the TD group to have influenced the results.

The study was approved by the ethical committee at ALYN hospital (protocols 030-20/044-21), according to the Declaration of Helsinki. The ethics committee waived the requirement for informed consent as this was a retrospective analysis, and all data were fully anonymized before the beginning of the study.

### Data collection

Participant characteristics were extracted from the hospital's electronic medical records. GMFCS level was assessed to define the functional level for each individual in the CP cohort.

**Table 1. Participants characteristics.**

| | | CP (n = 23) | | TD (n = 16) |
|---|---|---|---|---|
| | | **First assessment** | **Second assessment** | |
| Sex (Male/Female)* | | 17/6 | - | 6/10 |
| Age (Years)^ | | 9.0±2.5 | 10.6±2.5 | 9.8±2.7 |
| DWS (m/s)^ | | 0.25±0.07 | 0.25±0.08 | 0.41±0.03 |
| Body mass (kg)^ | | 31.9±14.5 | 39.3±17.1 | 36.1±14.0 |
| Functional mobility scale | 5 meters# | 5 [4–6] | 5 [2–6] | - |
| | 50 meters# | 5 [2–6] | 5 [2–6] | |
| | 500 meters# | 5 [0–6] | 5 [0–6] | |
| Previous orthopaedic intervention | Botulinum toxin * | 5 | - | - |
| | Soft tissue* | 8 | | |
| | SDR* | 1 | | |
| GMFCS level (I/II/III)* | | 9/13/1 | - | - |
| Uni/bilateral involvement* | | 12/11 | | |
| More involved side (Left/Right)* | | 9/14 | | |

Values are:

*numbers,

^mean±SD,

#median [minimum-maximum].

CP- cerebral palsy, TD- typically developing, DWS- dimensionless walking speed, SDR- selective dorsal rhizotomy, GMFCS- gross motor function classification system.

In order to evaluate changes in walking ability, the Functional Mobility Scale (FMS) was assessed. Both GMFCS level and FMS were scored by the physiotherapist conducting the gait analysis, and were recorded in the gait analysis report. A GRAIL system was used for the gait analysis (Motek, the Netherlands), consisting of: 1) an instrumented treadmill with two embedded force plates sampling at 1000 Hz (Motek, the Netherlands), 2) 10 optical motion tracking cameras sampling at 100 Hz (Vicon-UK, Oxford UK), 3) a 16-channel surface electro-myography (EMG) system sampling at 2000 Hz (Delsys Inc., Natick, MA, USA) and 4) a 180-degree screen and projection system (Motek, the Netherlands). The human body model-2 marker protocol was used to acquire gait kinematics (Motek, the Netherlands) [23, 24]. To measure muscle activity, EMG electrodes were placed on the following muscles, bilaterally: 1) gluteus medius, 2) medial hamstrings, 3) rectus femoris, 4) vastus lateralis, 5) medial gastroc-nemius, 6) soleus, 7) tibialis anterior and 8) peroneus longus. The electrodes were placed according to the SENIAM guidelines [25]. The data from the peroneus longus muscle presented very inconsistent activations patterns, possibly as a consequence of the small size of this muscle and its close proximity to the neighboring muscles (inter-muscular cross talk). The inclusion of this muscle in the synergy analysis could introduce an uncertain level of variability, potentially influencing the data. As such, this muscle was excluded from the study, resulting in seven muscles per limb. All markers and sensors were applied by experienced clinicians following standardised methods, using double sided-tape and robustly secured with micropore tape to reduce movement artefacts.

Individuals that are regular users of ankle-foot orthoses were usually assessed both barefoot and with their orthoses. However, for this study, only data from the barefoot condition were included. The participants walked on the treadmill whilst a virtual reality environment was projected onto the screen. This has been shown to engender the perception of an experience similar to over-ground walking [26]. All participants walked at a self-selected comfortable fixed speed, determined according to feedback from the participants and their families, following a minimum of six minutes walking to provide a period of familiarisation to walking on a treadmill [27, 28]. During this time, the EMG data were checked to ensure clear phasic activations or the presence of artefacts. If necessary, the sensor location was modified. Participants who ordinarily use assistive devices held onto the handrails for support while walking. Following the familiarisation period, a short break was provided, if requested. The participant then continued walking until a minimum of 50 consecutive gait cycles were acquired, per leg.

The timing of the second gait analysis was according to the recommendation of the referring medical team, and followed the same acquisition protocol as the first measurement. Both the first and second assessment protocols were identical, including the use of handrails by those participants who needed to do so. The only variable that changed was if the participant chose to walk at a different self-selected comfortable walking speed. No adverse events were reported during the assessments and none of the participants complained of discomfort when walking barefoot on the instrumented treadmill.

Moreover, it should be noted that the participants continued to receive their usual clinical treatment between assessments, including physical therapy (often with gait training that may involve treadmill walking), occupational therapy, hydrotherapy, and other relevant non-surgical interventions.

## Data analysis

In the software Nexus (version 2.8, Vicon-UK, Oxford UK), an algorithm was used to automatically detect and insert "foot strike" events throughout the gait trial, based on the vertical ground reaction force data and the locations of the second metatarsal and heel markers. All

events were visually inspected and manually adjusted, if necessary. A minimum of 30 strides per leg were segmented for each individual [29], except for three participants where, due to poor quality data, 10–15 strides were used. Poor quality EMG data consisted of artifacts that remained after filtering, or incomplete data set of EMG (i.e., missing muscles). To minimize any intrinsic measurement error, the number of analysed strides was equal for both assessments [29].

**Muscle synergy analysis.** All EMG signals were analysed using custom MATLAB scripts in MATLAB 2021b (Mathworks Inc., Natick, MA, USA). The signals were filtered using a sixth order Butterworth high-pass filter with a cut-off frequency of 20Hz, followed by a 50Hz notch filter. The EMG signals were then rectified and smoothed with a fourth-order Butterworth lowpass filter with a 10Hz cut-off frequency. The EMG signals were split into strides based on the foot strike events. Each stride was resampled to 101 data points, representing 0% to 100% of the gait cycle. The concatenated signal was then normalized to its mean amplitude. Non-negative matrix factorisation was used to calculate the total variance in EMG activity accounted for by the different synergy solutions [21]. The total variance in EMG activity accounted for by one synergy ($tVAF_1$) was calculated for each participant from each assessment [6, 21, 30]. Previous studies have revealed that the $tVAF_1$ is reliable between visits (1–42 days between assessments) [31, 32]. Lastly, the $tVAF_1$ was transformed to a z-score based on the data of the TD participants to create the walking dynamic motor control index (walk-DMC) [12]. Walk-DMC is a measure whereby 100 is the mean for TD participants and a change in 10 points is one standard deviation (SD) [12]. A reduction in Walk-DMC suggests less complex SMCg. The Walk-DMC is the primary outcome measure of this study.

**Kinematics.** Gait deviations were assessed using the Gait Profile Score (GPS) [33]. The GPS is a raw value based on the root mean square distances between 15 lower limb joint angle trajectories of an individual and their averages with respect to TD participants data [33]. The GPS was chosen as it can be decomposed to provide the Gait Variable Score (GVS), an index that measures the deviation from the TD reference data for each of the kinematics variables: anterior/posterior pelvic tilt, hip flexion/extension, knee flexion/extension, ankle dorsiflexion/plantarflexion, up/down pelvic obliquity, hip abduction/adduction, pelvic internal/external rotation, hip internal/external rotation and foot progression. The GVS values do not provide information regarding the direction of change, only the deviation from the TD values. Before calculating the GPS and GVS values, experienced clinicians performed a quality and consistency check of the kinematic data in the software Gait Offline Analysis Tool 4.2 (Motek, the Netherlands), to remove poor quality strides. Poor quality kinematic data consisted of strides with large deviations (i.e., looking to the side and inducing excessive joint rotation), or artifacts due to irretrievable gaps in marker data. The data from the remaining strides were then exported and processed in a custom python script (Spyder version 5.4.0) to provide the GPS and GVS values. In children with CP, the minimal clinically important difference in GPS value is 1.6˚ [34]. The GVS values can also be displayed on a bar chart known as the Movement analysis profile [33].

When analyzing muscle synergies and kinematics for both legs together (bilateral analyses), the foot strike events of the right leg were used to split the signals into gait cycles for both sides [33]. Based on preliminary testing on a subgroup of the participants, there was little influence on the bilateral walk-DMC value when segmenting the gait trials according to either the more or less involved leg. For the CP cohort, when muscle synergies and kinematics were extracted for the more involved leg, the foot strike events of that leg (the most involved) was used to split the signals into gait cycles. More or less involved leg was determined according to the clinical physical examination conducted as part of the clinical gait analysis protocol used in our lab. The selected strides for extracting the walk-DMC and GPS/GVS were not always identical. For

example, there were strides that contained good quality EMG data, but not kinematic data. These strides were only used for extraction of the walk-DMC, and not for the calculation of the GPS or GVS. However, due to the large number of strides that were included it is not anticipated that this would lead to a change in the results.

Along with the changes in functional level (FMS), changes in body mass and dimensionless walking speed (DWS) [35] between the two gait assessments were extracted and considered as confounding factors, as they may have an effect on an individual's gait pattern [12, 31, 36, 37].

### Sample size calculation

Sample size was calculated based on previous findings, suggesting a mean absolute change in $tVAF_1$ of 4.2 ± 3.1% between gait analysis visits [32], and a maximum acceptable error of 2% in the $tVAF_1$ values between assessments [29]. As such, an a-prior analysis with the assumption for first-class error of 5% and a required power of 80% revealed a minimum sample size of 20 participants (WINPEPI ltd) [38].

### Statistical analysis

The assumption of normal distribution was assessed by the Shapiro-Wilk test (p>0.050). The change in the different variables between the first and second assessment was evaluated by a Wilcoxon signed-rank test or paired sample t-test, as needed. Effect sizes were calculated using GPower V.3.1.7 (University of Kiel, Kiel, Germany). In order to rule out the influence of confounding factors, changes in functional level, body mass and walking speed were also examined. The relationship between the different variables were assessed using either a Spearman's rank or Pearson correlation coefficient. According to the number of hypotheses, statistical significance was set at p<0.025 (0.050/2) [39]. Statistical analyses were performed using SPSS version 27 (SPSS Inc., Chicago, IL, USA).

## Results

Participant characteristics in the first and second assessments are presented in Table 1. The mean time between the assessments was 590±202 days (minimum 193—maximum 992 days). There was no significant relationship between the interim period and age, whether the individuals with CP were younger or older.

In both assessments, most participants were regular users of ankle-foot orthoses (n = 21) and ambulated without assistive devices (n = 20). Most participants (n = 19) did not use handrails during the assessments; an initial analysis did not reveal any significant effect of using handrails on the results.

### Longitudinal changes in SMCg

Walk-DMC scores for the more involved leg and for both legs together, from both assessments, were normally distributed, and are presented in Table 2. No significant change was found in the mean walk-DMC score between the first and the second assessment. However, there was a large variation within group, whereby the walk-DMC of the involved leg increased in 11 participants and decreased in 12 participants (Fig 1a), whilst in the bilateral analysis, the walk-DMC increased in 15 participants and decreased in eight participants (Fig 1b).

With respect to the influence of potential confounding variables, there were no significant changes in the functional level, according to the FMS, and in the DWS (Table 1). However, body mass increased significantly in the second assessment by a mean of 7.4 kg (95% CI 5.6–9.1 kg; p<0.001) (Table 1). No significant or meaningful correlations were found between the

**Table 2. Muscle synergies parameters and GPS value in both assessments.**

| n = 23 | | First assessment | Second assessment | p-value | Effect size |
|---|---|---|---|---|---|
| Walk-DMC score^ | More involved leg | 81.7±15.4 | 80.7±12.2 | 0.746 | 0.03 |
| | Both legs together | 85.3±11.3 | 84.5±9.7 | 0.757 | 0.06 |
| GPS value#* | More involved leg | 9.6 [5.5–26.7] | 10.8 [4.5–25.2] | 0.133 | 0.21 |
| | Both legs together (total value) | 9.9 [5.7–25.5] | 10.7 [5.3–26.3] | 0.372 | 0.14 |

Values are:

^mean±SD,

#median [minimum-maximum].

Walk-DMC- the walking dynamic motor control index, GPS- gait profile score.

*n = 18 in the second assessment

changes in functional level, body mass or DWS to the changes in walk-DMC in the more involved leg or both legs together. In addition, no significant correlations were found between age at the first assessment or the time between assessments with changes in walk-DMC for either the more involved leg or both legs together. Controlling for topographical limb involvement (bilateral/unilateral) or previous orthopedic intervention did not influence the results.

## The relationship to gait deviations

All 23 participants had good quality kinematic data in the first assessment, however due to technical issues, only 18 participants had good quality kinematic data during the second assessment. The GPS values for the more involved leg and both legs together, in both assessments, were not normally distributed, and are presented in Table 2. The Movement analysis profile for the more involved leg is presented in Fig 2. No significant change was found in the median GPS or GVS values between the first and second assessment. However, there was a large variation within group. The GPS value in the more involved leg increased by more than the minimally clinically important difference of 1.6° in two participants, decreased by more than 1.6° in eight participants, and was below the minimally clinically important difference in the remaining eight participants. No significant correlations were found between the change in the walk-DMC with the GPS value in the more involved leg, or the total GPS value.

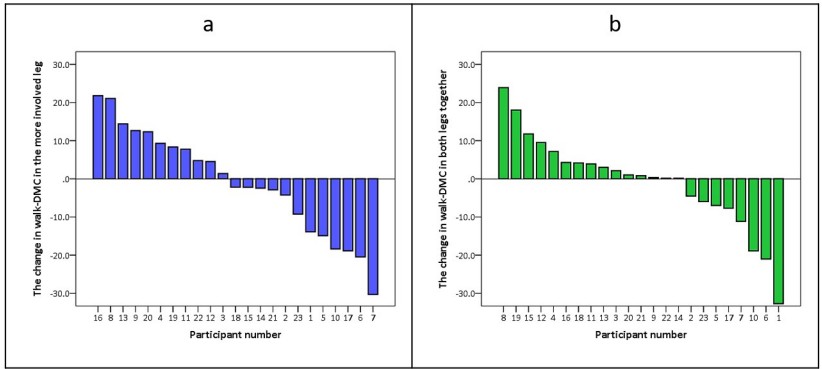

**Fig 1. Overview of change in walk-DMC scores between the two assessments.** In both figures, participant numbers have been ordered from greatest improvement to greatest deterioration in walk-DMC scores, with the order of participant numbers differing between the two figures. Walk-DMC- the walking dynamic motor control index.

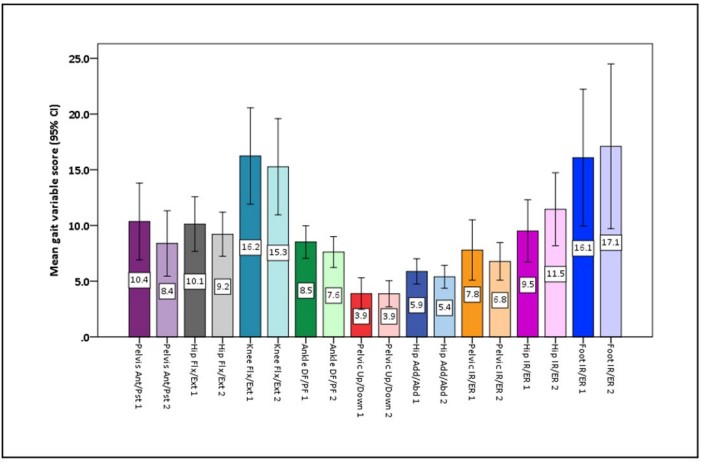

**Fig 2. Movement analysis profile for the more involved leg (n = 18).** The numbers within the bars represent the mean and the lines are the 95% confidence intervals. Dark colours—first assessment, light colours—second assessment. Ant- anterior, Post- posterior, Flex- flexion, Ext- extension, DF- dorsiflexion, PF- plantarflexion, Add- adduction, Abd- abduction, IR- internet rotation, ER- external rotation.

However, with respect to the GVS, a significant moderate negative correlation was found between the change in walk-DMC to the change in hip flexion/extension GVS value in the more involved leg ($r_p$ = -0.56, p = 0.017; Fig 3a). These findings indicate that when the walk-DMC value increased (improved SMCg complexity), there was less deviation in hip flexion/extension kinematics. Moreover, a significant moderate positive correlation was also found between the change in walk-DMC and the hip internal/external rotation GVS value in the more involved leg ($r_p$ = 0.65, p = 0.004, Fig 3b). These findings indicate that when the walk-DMC increased, (improved SMCg complexity), there was more deviation in hip internal/external kinematics. An overview of the changes in the hip flexion/extension and hip internal/external rotation GVSs between the assessments is presented in Fig 4.

No other correlations were found between the change in walk-DMC to the GVS values.

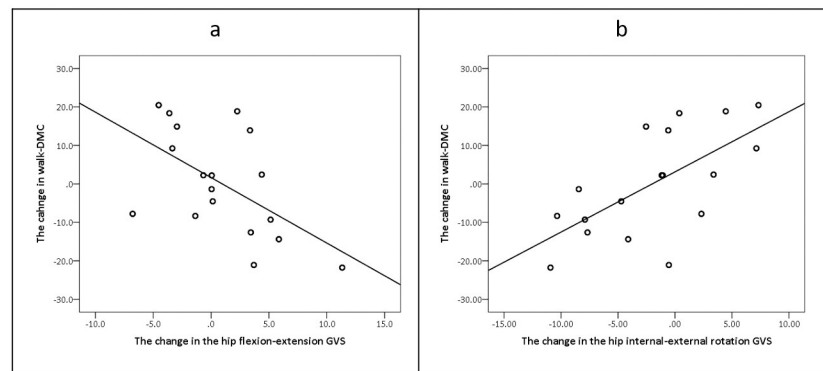

**Fig 3. The relationship between the changes in the walk-DMC to the hip flexion/extension GVS (a), the relationship between the changes in the walk-DMC to the hip internal/external rotation GVS (b).** All relationships were in the more involved leg. Walk-DMC- the walking dynamic motor control index, GVS- gait variable score.

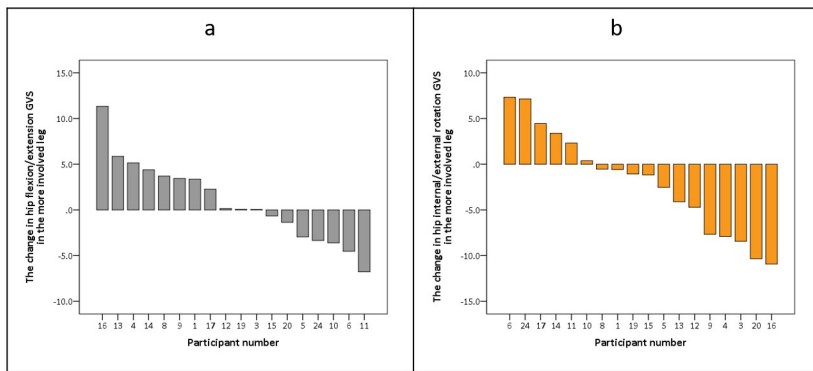

**Fig 4. Overview of change in hip flexion/extension and hip internal/external rotation GVS values, between the two assessments.** In both figures, participant numbers have been ordered from greatest improvement to greatest deterioration in the GVS value, with the order of participant numbers differing between the two figures. GVS- gait variable score.

## Discussion

This study evaluated longitudinal changes in SMCg in individuals with CP, and whether this related to changes in gait deviations. There was no change in walk-DMC between assessments, confirming the first hypothesis. However, high variability was found between participants for the more involved leg. The change in Walk-DMC score was found to relate to changes in hip flexion-extension and internal-external rotation GVSs between assessments. This partially confirmed the second hypothesis that changes in SMCg will relate to changes in gait deviations.

Prior to evaluating changes in SMCg over time, we explored the relationship between age and walk-DMC at the time of the first assessment, without any significant findings. This is in line with a previous study in a very large cohort [12]. Additionally, Cappellini et al also demonstrated very limited age-related changes in muscle synergies in children with CP, based on a single assessment in a wide age range of participants [13]. It has also been shown that SMCg in individuals with CP can often resemble the synergies observed in much younger TD children, highlighting not only the delay but also stagnation in SMCg development [40]. However, as previously reported by Bekius et al. in young children with CP [11], we did notice a large variability in how SMCg changes over time between individuals, when assessed by the walk-DMC. In the analysis of the more involved leg, 12 participants presented with an increase in SMCg complexity and 11 participants a decrease in SMCg complexity (Fig 1). This variability can explain the minimal change in the mean walk-DMC value between assessments (resulting in a mean change around one value), reinforcing the importance of individual analysis with respect to SMCg.

Although we did not identify a relationship between changes in walk-DMC and the GPS, there was a relationship between changes in walk-DMC and the sagittal and transverse plane hip GVS of the more involved leg. In a previous study, it has been suggested that level of SMCg might be indicative of the capacity of the central nervous system to selectively activate muscles during a dynamic task. This capacity might also affect gait kinematics, with more alterations across multiple joints when SMCg is more impaired [21]. However, a direct association between a gait pattern and level of SMCg was not found in this study, similar to our results for the GPS. However, we did find that the sagittal and transverse plane hip kinematics might be associated with level of SMCg. The importance of sagittal hip position in regulating motor

control of walking in cats was reported over 20 years ago [41]. However, whether hip position actually plays an important role in regulating human walking or what the mechanism behind this association may be, remains undetermined.

In this study, all participants walked with an increase in hip flexion throughout the gait cycle during the first assessment. In the second assessment, those participants that presented with a relative increase towards hip extension in the more impaired leg (normalisation of hip kinematics), also demonstrated an increase in walk-DMC score. This occurred without any significant changes in sagittal plane pelvic tilt or walking speed. An increase in hip extension during the gait cycle can result in a more extended posture and posterior shift of the center of mass, optimising the alignment of the ground reaction force around the hip, knee and ankle [42]. This could alter the activation timing of the hamstring muscles (as hip extensors) and quadriceps muscles (as knee extensors) during the stance phase, resulting in less lower limb muscle co-activation, and, a better walk-DMC score. In addition. an increase in hip extension and limb loading has been shown to prolong the stance phase period in infant stepping, indicating that sensory information from the hip may play an important role in the regulation of gait [18].

Regarding the transverse plane hip ROM, a change in hip internal/external rotation was correlated with a change in walk-DMC. Further analysis revealed that unlike hip flexion/extension, the direction of the change was highly variable between participants. This may be a consequence of changes in femoral anteversion and instability around the foot, two clinical findings that influence the lever arm of the external hip rotator muscles [42, 43] and may influence muscle activity and SMCg. Further study is required to determine the clinical relevance of the finding of changes in hip internal/external rotation GVS with walk-DMC.

Interestingly, this study only found relationships between changes in walk-DMC and hip kinematics, despite altered kinematics also observed at the pelvis, knee and ankle. However, previous studies have demonstrated the importance of the hip position on modifying the gait cycle. In individuals with CP, altered hip muscle activity, including changes in the firing rate, recruitment and synchrony of motor units, were found to contribute to muscle fatigue and decreased biomechanical efficiency [44]. Additionally, afferent input from hip joints and load receptors, play a crucial role in the generation of locomotor activity [17]. Therefore, the reduction in muscle co-activation reflected by an increase in walk-DMC score, may be a consequence of both better motor control and biomechanical demands, leading to a more optimal gait pattern.

Along with the findings of previous studies [10, 11, 21], the current study brings into question the fixed nature of SMCg in individuals with CP, beyond the usual considered timeframe related to corticospinal tract development (around the age of two years). Our results reinforced the notion that SMCg is highly variable between individuals [6, 21], and suggests that, to some extent, it is related to changes in joint kinematics. The development of SMCg may be individualized and dependent on internal (e.g., location and severity of the brain lesion) and external factors (e.g., treatment type and duration) [21]. Further investigation should examine the longitudinal changes in muscle synergies and gait function from the stage of independent ambulation, through to adolescence.

Regarding the limitations of this study, it should be acknowledged that the sample size, although sufficiently powered, is modest. Therefore, while the findings of this study may be valuable and informative, they should be interpreted with caution. Second, the participants were referred for a gait analysis due to a clinical question. Thus, there may be a bias with respect to the heterogenous convenience sample of participants included in the analysis, reducing the generalisability of these results. Third, the interim period between gait assessments in the study was variable. Although no correlation was identified with walk-DMC

scores, future studies may benefit from having follow-up evaluations according to a standard-ised period of time. Lastly, this study evaluated the relation between walk-DMC and GPS/GVS, the latter being a measure of kinematic deviation. Future investigations should consider the addition of a kinetic measure, such as the GDI-kinetic [45], which may have more relation with muscle activations.

This study has several strengths that enhance the validity and significance of its findings. Firstly, it incorporates an objective evaluation of muscle activity and gait kinematics. Second, a minimum of 30 strides per leg were included for most participants. This method not only strengthens the validity of our study results but also enhances the reliability of the data collected. Additionally, to the best of our knowledge, this study is the first to evaluate longitudinal changes in SMCg in a group of individuals with CP without any surgical interventions between assessments (including botulinum toxin injections). It also examines whether changes in SMCg are associated with variations in gait deviations. As a result, this study provides valuable insights into the natural progression of SMCg during gait, previously unreported.

## Conclusion

While studying a group of individuals with CP, it was observed that on average there were no significant longitudinal changes in SMCg. However, there was considerable variability between individuals, which could be attributed to changes in hip joint kinematics in the sagittal and transverse planes (flexion-extension and internal-external rotation) of the more involved leg. This suggests that there remains a capacity of the central nervous system to alter its control strategy of walking, with changes in joint kinematics as a result of that strategy. No relationships were found with age, body mass, or walking speed. Further research should evaluate specific interventions to improve hip kinematics during the stance phase of gait and its influence on SMCg, and assess the effect on the longitudinal changes in muscle synergies from early childhood to adolescence.

## Acknowledgments

We would like to thank the clinical team at the ALYN motion analysis laboratory of for the assistance with collection and extraction of the data for this study.

## Author Contributions

**Conceptualization:** Gilad Sorek, Simon-Henri Schless.

**Data curation:** Gilad Sorek, Itai Schurr, Simon-Henri Schless.

**Formal analysis:** Gilad Sorek, Marije Goudriaan, Simon-Henri Schless.

**Funding acquisition:** Marije Goudriaan, Simon-Henri Schless.

**Investigation:** Gilad Sorek, Itai Schurr, Simon-Henri Schless.

**Methodology:** Gilad Sorek, Marije Goudriaan, Simon-Henri Schless.

**Project administration:** Simon-Henri Schless.

**Resources:** Gilad Sorek, Itai Schurr, Simon-Henri Schless.

**Software:** Marije Goudriaan, Simon-Henri Schless.

**Supervision:** Simon-Henri Schless.

**Validation:** Gilad Sorek, Marije Goudriaan, Simon-Henri Schless.

**Visualization:** Gilad Sorek.

**Writing – original draft:** Gilad Sorek, Simon-Henri Schless.

**Writing – review & editing:** Gilad Sorek, Marije Goudriaan, Itai Schurr, Simon-Henri Schless.

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
