## [Decision Letter · Decision Letter 0]

14 Mar 2023

PONE-D-23-03502A longitudinal analysis of selective motor control during gait in individuals with cerebral palsy and the relation to gait deviationsPLOS ONE

Dear Dr. Sorek,

Thank you for submitting your manuscript to PLOS ONE. After careful consideration, we feel that it has merit but does not fully meet PLOS ONE’s publication criteria as it currently stands. Therefore, we invite you to submit a revised version of the manuscript that addresses the points raised during the review process.

We look forward to receiving your revised manuscript.

Kind regards,

Roxana Ramona Onofrei, PhD, MD

Academic Editor

PLOS ONE

Journal Requirements:

Additional Editor Comments:

The manuscript presents interesting results. However, there are some methodological concerns that could have a negative impact on the results:

- why have the authors choose only 16 typical developed children for control reference values? How was this sample size established? They should be at least age-matched. There is a difference between the age range in CP participants and TD ones. What about these children characteristics - height, weight?

- walking barefoot on the treadmill was comfortable for children? Could this have affected the gait analysis? In the results section (first paragraph) it is mentioned that in both assessments most participants were regular users of ankle-foot orthoses. Were they wearing the orthoses during gait assessments?

- what about the difference between CP participants that used handrails for support while walking and those who did not?

Reviewers' comments:

Reviewer's Responses to Questions

**Comments to the Author**

1. Is the manuscript technically sound, and do the data support the conclusions?

Reviewer #1: Yes

Reviewer #2: Partly

2. Has the statistical analysis been performed appropriately and rigorously? 

Reviewer #1: Yes

Reviewer #2: Yes

3. Have the authors made all data underlying the findings in their manuscript fully available?

Reviewer #1: Yes

Reviewer #2: Yes

4. Is the manuscript presented in an intelligible fashion and written in standard English?

Reviewer #1: Yes

Reviewer #2: Yes

5. Review Comments to the Author

Reviewer #1: Dear Authors,

Thank you very much for submitting your manuscript to PLOS ONE. The readers of the PLOS ONE will be interested in the topic of present study. I have some concerns about your study. I indicated my comments for each section of the manuscript.

Abstract

Keywords

- Mesh terms do not match the developmental delay and muscle synergies. Please double-check your keywords using MeSH.

Introduction

o Please update the authors name in the reference number one. (eg: Damiano, DiL not correct)

- Introduction is well written. The authors used current references from the literature.

Method

- Assessments and sample size were described in detailed. But authors should indicate the calculation of sample size. How did you decide to involve 16 health peers into the study?

Results

- You should present the sociodemographic characteristics of children without CP (typically developing peers) such as age, gender, body mass etc.

- Are there any adverse events in the groups during the assessments?

- You should include the effect sizes for differences within group differences in Table 2.

Discussion

- Limitations are more comprehensive. I think sample size is one of the limitations of the study.

- Please also include the strengths of the study. For example, using objective evaluation methods are one of the strengths of the study.

Reviewer #2: General comments

This is an interesting study. The sbmitted manuscript is generally well thought-through and adresses relevant and novel topics. However, I have some major concerns and I would advice against publication of the study in it’s current form. My concerns are mainly about the interpretation and conclusions that the authors draw from the study results. I believe that more nuance would be in place and too much emphasis is put on some minor results.

Already in the abstract the authors claim quite strongly that ‘individuals with CP demonstrated longitudinal changes in SMCg when evaluated on an individual level’ and ‘add to the evidence that SMCg in individuals with CP may continue to change over time’. Unfortunately, this is not supported by a statistical change in SMCg in the results. Although I am generally in favor of reporting individual results in addition to group statistics, especially in a heterogeneous population like individuals with CP, the authors frame it as an improvement on individual level. While in fact, the only valid conclusion would be that they found no change in SMCg. Of course, further exploration into the reasons for the absence of any significant change is valuable, but the way it is presented now is confusing and in some sections even misleading.

Secondly, while the authors put a lot of emphasis on the outcomes of the hip joint, this data is not presented clearly in the results. Presentation of data supporting their conclusions is missing (see specific comments below). Moreover, I have some methodological concerns on the outcomes of the hip joint which are listed below in detail.

My final general concern is about the discussion paragraphs. This section should be revised thoroughly as in it’s current form, it lacks a critical evaluation of the findings and implications of the methodological choices made by the authors.

Specific points are listed below.

Introduction

Page 4 – line 4: Hypotheses on significant changes in SMCg between assessments versus changes on an individual level. It is unclear what the authors mean by ‘changes on an individual level’. Assuming that a statistical design is used to test within-subject changes over time, doesn’t that represent individual changes? Please rephrase.

Materials and methods

Participants

Page 5 – line 8: The age range of 4-18 years is quite large. Especially in combination with the time range between assessments (results, page 10: 193-992 days). How was this distributed over the group? I.e. did younger children have more/less days in-between assessments? Could the authors also provide information on the reason why children were seen for gait analysis?

Data collection

Page 5 – line 24: How was GMFCS scored? Did the authors expect this outcome to change, as it is used as an outcome for function?

Page 6 – line 7-8: Human body model-2: Suggestion to use the reference of vd. Bogert et al., 2013 for HBM or Falisse et al., 2018 / Flux et al., 2020 for description of the updates in the updated version.

Line 17: Did the children truly walk barefoot on the treadmill? Or were non-supporting flexible shoes supplied as commonly used on the treadmill for safety and comfort?

Lines 23-13: Removing the signal of peroneus longus from analysis. Not sure why cross-talk of the peroneus would be a problem when conducting synergy analysis. Isn’t synergy analysis about activated groups of muscles together, instead of evaluating the pattern of individual muscles?

Lines 19-20: Was the treadmill used in a self-paced mode or fixed?

Line 21-22: Sufficient adaptation to walking on a treadmill: Please rephrase, as the term ‘sufficient’ is vague. It questionable whether 6 minutes are truly ‘sufficient’ in this population, as the study by Meyer 2019 was conducted on healthy adults and did not include EMG measurements.

Page 7 – lines 3-5: Was the same walking speed imposed during the second assessment?

Data analysis

Page 7 – lines 11-14: Was the visual quality check for EMG data done simultaneously for kinematics as described on page 8, lines 22-24? In other words, were the same strides included for these analyses? Or could this differ?

Page 9 – line 3: I assume the authors mean ‘the minimal clinically important difference’, instead of ‘mean important change’. The rest of this sentence is also unclear, please rephrase.

Page 9 – lines 8-14: I don’t understand what is described in this paragraph? Which leg was used for analysis? It is unclear what is meant by ‘For the walk DMC, this was based on the less-involved leg to provide interparticipant consistency’. Please rephrase.

Page 9 – line 16: Typo, abbreviation dimensionless walking speed should be DWS.

Results

Page 10-11 , Table 1: Please indicate in the title that this table presents characteristics of participants with CP (not patients).

Table 1: Can the authors provide information on the type of orthopaedic interventions? As a percentage of 49% is quite high. Furthermore, it should be explained more clearly what the numbers in the table represent. Please indicate per outcome whether it concerns a number, mean, SD or range (min-max). The age range in the Table does not match the range 4-18 described on page 5.

Page 11: Longitudinal changes in SMCg: Were data normally distributed? I.e. do the results represent outcomes of parametric or non-parametric tests?

Again, splitting outcomes in ‘on group level’ and ‘on the individual level’ is confusing, as the used statistics should account for inter-individual differences. The authors found no statically significant changes in SMCg. The fact that there are large inter-individual differences is nice to present as additional information, but it should not be presented as ‘changes on individual level’.

Page 11-12 - Table 2: GPS value- total value also refers to both legs together?

Page 12 – line 2: Personally, I would not expect any change in GMFCS, even if there would be a change in functional level as this outcome is probably not sensitive enough to detect these changes. Actually, it is meant to be robust against (natural/expected) functional changes during growth and development. Reconsider using this as an outcome of ‘change on a functional level’.

Page 13 – line 1-2: Why are no results presented of the MAP-score for both assessments in Figure 2? This is required to place the findings of the hip GVS in perspective. At this point, I sincerely doubt whether the significant relation between the change in hip joint GVS and walk-DMC is not just a lucky shot.. Results showing mean +/-SD for both assessments should be added.

Page 13 – lines 1-2: Using the terms GVS and MAP alternately throughout the paper is confusing, especially for readers who are less familiar with these concepts. It would benefit readability to choose one of them. I have a comparable issue with the terms SMCg and DMC, as both abbreviations basically represent the same phenomena. Sticking to one term makes it a lot easier to read.

Page 13 – line 15 to page 14 line 5: Remark on using the MCID values for GVS on the current dataset. The values of 1.5 deg and 0.6 deg for the hip joint reported by Baker et al are based on the PlugInGait model. However, the underlying biomechanical calculations for HBM are different, using global optimisation to model markers. This may especially affect hip rotation, which is measured with just one lateral marker on the thigh.

Discussion

Page 14 – line 9: ‘confirming the first hypothesis’. What was the first hypothesis? Please repeat the hypothesis or rephrase this sentence (same for the second hypothesis in line 12).

Line 9: As mentioned above. Please drop the words ‘on individual level’.

Page 14 – lines 10-11: I assume the authors mean that the change in walk-DMC was related to changes in hip flexion – extension … GVS? As mentioned above, consider to use just one term for reporting the outcomes of motor control.

Page 15 – lines 1-2: A conclusion of this paragraph is missing. What is the point of repeating the finding that the authors found no change in SMCg? What is their interpretation?

Lines 3-9: As described above. I continue to have a problem with reporting this finding as improvement on an individual level. In my opinion, it could also mean that (1) there is no change over time (2) the variability could also raise questions on the robustness of this outcome.

Lines 10-13: See above on my questions regarding these outcomes of the hip joint. More extensive presentation of the results is required to support this conclusion.

Lines 13-17: This is quite speculative and should rephrased. See previous comment.

Lines 22-24: Could habituation to treadmill walking also affect these result, leading to a more upright posture during the second assessment? How many children used the handrail for support? And did this change between assessments?

Page 16 – lines 4-5: This is no causal effect, as muscle co-contraction and motor control quantified by muscle synergies capture different phenomena. Please rephrase.

Page 16 – lines 7- to end: Again, I have doubts on the importance of the results in the hip joint. Especially for quantification of hip internal-external rotation using the HBM model. This paragraph of the discussion is speculative and puts (in my opinion) too much emphasis on the results of the hip rotation. In contrast, a critical discussion of the findings of this paper and the limitations of the methods used is missing at this point.

Page 18 – lines 6-11: Including different strides for kinematics and EMG. This should be included in the results sections or methods. Moreover, it is not clear what point the authors try to make by saying that they therefore ‘did not evaluate their interaction’.

Lines 14-16: Is it known whether participants walked on the GRAIL or another treadmill system during the interim period in-between assessments? See my question about habituation. Were children seen by other therapists or did they have any other therapy or training that may affect their gait?

6. PLOS authors have the option to publish the peer review history of their article (what does this mean?). If published, this will include your full peer review and any attached files.

Reviewer #1: **Yes: **Ozgun Kaya Kara

Reviewer #2: No

---

## [Author Response · Author response to Decision Letter 0]

26 Apr 2023

The responses to the reviewers are in the "response to the reviewers" letter that was attached

---

## [Decision Letter · Decision Letter 1]

12 Jul 2023

A longitudinal analysis of selective motor control during gait in individuals with cerebral palsy and the relation to gait deviations

PONE-D-23-03502R1

Dear Dr. Sorek,

We’re pleased to inform you that your manuscript has been judged scientifically suitable for publication and will be formally accepted for publication once it meets all outstanding technical requirements.

Kind regards,

Roxana Ramona Onofrei, PhD, MD

Academic Editor

PLOS ONE

Additional Editor Comments (optional):

The manuscript has been significantly improved.

Reviewers' comments:

Reviewer's Responses to Questions

**Comments to the Author**

1. If the authors have adequately addressed your comments raised in a previous round of review and you feel that this manuscript is now acceptable for publication, you may indicate that here to bypass the “Comments to the Author” section, enter your conflict of interest statement in the “Confidential to Editor” section, and submit your "Accept" recommendation.

Reviewer #1: All comments have been addressed

2. Is the manuscript technically sound, and do the data support the conclusions?

Reviewer #1: Yes

3. Has the statistical analysis been performed appropriately and rigorously? 

Reviewer #1: Yes

4. Have the authors made all data underlying the findings in their manuscript fully available?

Reviewer #1: Yes

5. Is the manuscript presented in an intelligible fashion and written in standard English?

Reviewer #1: Yes

6. Review Comments to the Author

Reviewer #1: The authors have made excellent modifications which has resulted in substantial improvement to the clarity of the manuscript.

7. PLOS authors have the option to publish the peer review history of their article (what does this mean?). If published, this will include your full peer review and any attached files.

Reviewer #1: No

---

## [Editor Report · Acceptance letter]

21 Jul 2023

PONE-D-23-03502R1 

A longitudinal analysis of selective motor control during gait in individuals with cerebral palsy and the relation to gait deviations 

Dear Dr. Sorek:

I'm pleased to inform you that your manuscript has been deemed suitable for publication in PLOS ONE. Congratulations! Your manuscript is now with our production department. 

Kind regards, 

on behalf of

Dr Roxana Ramona Onofrei 

Academic Editor

PLOS ONE